# Effects of Sunflower Hulls on Productive Performance, Digestibility Indices and Rumen Morphology of Growing Awassi Lambs Fed with Total Mixed Rations

**DOI:** 10.3390/vetsci8090174

**Published:** 2021-08-30

**Authors:** Abdualrahman Salem Alharthi, Hani Hassan Al-Baadani, Mohammed Abduh Al-Badwi, Mutassim Mohammed Abdelrahman, Ibrahim Abdullah Alhidary, Rifat Ullah Khan

**Affiliations:** 1Department of Animal Production, College of Food and Agriculture Sciences, King Saud University, Riyadh 11451, Saudi Arabia; abdurahman@ksu.edu.sa (A.S.A.); halbadani@ksu.edu.sa (H.H.A.-B.); malbadawi@ksu.edu.sa (M.A.A.-B.); amutassim@ksu.edu.sa (M.M.A.); 2College of Veterinary Sciences, Faculty of Animal Husbandry and Veterinary Sciences, The University of Agriculture, Peshawar 250000, Pakistan; rukhan@aup.edu.pk

**Keywords:** agricultural byproduct, carcass traits, digestibility, Awassi lamb, productive performance, rumen morphology, sunflower hulls

## Abstract

Forty-eight growing Awassi lambs were used in a 70-day trial to investigate the effects of different levels of dietary sunflower hulls (SFH) on growth, rumen morphology, fiber digestibility and meat characteristics of lambs. Animals were randomly allocated to 4 groups with 3 replicates of 4 lambs each. The diet was composed of total mixed ration (TMR) without SFH (control group), and the TMR diet supplemented with SFH at a level of 5% (SFH5), 10% (SFH10) and 15% (SFH15). Lambs in the treatment groups had greater BW changes (*p* = 0.04) and ADG (*p* = 0.04) than the lambs in the control group. Intake of dry matter, acid detergent fiber (ADF) and neutral detergent fiber (NDF) were also significantly (*p* < 0.05) higher in SFH15 compared to SFH10. Digestibility of ADL and empty stomach weight were also significantly (*p* < 0.05) higher in SFH10 and SFH15, respectively. Cooking loss, blood total cholesterol and total protein decreased significantly (*p* < 0.05) in SFH15. Ruminal lightness (L) and yellowness (b) also increased significantly (*p* < 0.05) in SFH15. We concluded that the TMR diet supplemented with up to 15% SFH improved weight gain, digestibility, meat cooking loss and rumen color in Awassi lambs.

## 1. Introduction

In the oil extraction industry, a large quantity of residues are produced that can be used as animal food for ruminants as a protein source, aiding progress towards a sustainable production system and the goal of a circular-economy model [1]. Sunflower hulls (SFH) are the byproducts obtained after a crushing operation of sunflower seeds. The nutrient composition of SFH varies considerably depending on the type of seed [2]. In general, SFH contained approximately 5–7.1% crude protein (CP), 3–11% fat and 42.4–56% acid detergent fiber [3,4]. Sunflower seeds are mostly used for the extraction of oil; however, they are also a valuable source of protein and fat for ruminants [1,5].

SFH are an effective source of dietary fibers, which can meet the need for roughage. They have a positive effect on feed intake and fiber digestibility when used for up to 35% of the diet in dairy cows [6] and are acceptable at 25% of diets for lactating cows [7]. In sheep, SFH addition at the level of 30 g/kg in diets had no negative effects on feed consumption and digestibility indices [8]. Alobre et al. [9] reported that SFH can be used for up to 20% of the diet of pregnant sheep with no negative effect on the nutritional composition of colostrum. Park et al. [2] reported that heifers fed rations containing 27% SFH had the highest weight gain (1.36 kg/day) compared with those fed with 50% SFH. In addition, lambs fed with SFH improved cis-9, trans-vaccenic acid, trans-11 octadecadienoic acid in muscle and fats tissue [10,11]. Several studies reported that the addition of certain amounts of insoluble fiber sources stimulated the development of the gastrointestinal tract (GIT) in broilers [12,13]. However, little is known on the effects of SFH on growth, GIT traits and nutrient digestibility in small ruminants. Therefore, the purpose of the current study was to determine the effects of SFH as a roughage source in rations on production performance, nutrient digestibility, rumen morphology and carcass characteristics for growing Awassi lambs.

## 2. Materials and Methods

### 2.1. Animals and Experimental Design

A total of 48 growing Awassi lambs (mean BW, 29.2 ± 1 kg; 4 months-old) were used in a 70-day trial. Animals were purchased from the local livestock market and then transported to the Experimental Station of Animal Production Department, College of Food and Agriculture Sciences, King Saud University, Riyadh. On the arrival day, lambs were immediately weighed, ear-tagged, vaccinated against clostridial diseases and treated for internal and external parasites. Thereafter, animals were randomly allocated to 4 groups with 3 replicates of 4 lambs each. The lambs were housed as a group, feeding in shaded pens. The average temperature and relative humidity were 35 °C ± 0.1 and 55% ± 3, respectively. All lambs were offered total mixed rations (WAFI, ARASCO, Riyadh, Saudi Arabia). The diet fully covered the requirements of NRC [14] recommendations and were offered twice daily at 08: 00 and 15: 00. Water was available *ad libitum*. On day 1 of the experimental period, lambs were randomly assigned to one of four dietary treatments consisting of total mixed ration (TMR) without adding SFH (control group, CON), a basal diet supplemented with SFH at a level of 5% (SFH5), 10% (SFH10) and 15% (SFH15) as shown in Table 1. Sunflower hulls were milled and mixed well at the given concentration in the respective groups. At the termination of the study (day 70), all lambs were first deprived of feed and water for 16 h and then slaughtered, using the Islamic method, to evaluate ruminal morphology, carcass characteristics and meat quality.

### 2.2. Measurements and Sampling

Feed from each treatment was sampled before the study and weekly during the study, and samples were frozen at −20 °C. At the end of the study, feed samples were pooled (5%) and analyzed for nutrient composition. Dry matter (DM) content was determined by drying samples in an oven at 100 °C for 24 h, and ash content was determined by incinerating samples at 550 °C for 3 h in a muffle furnace. Crude protein content was measured using an elemental analyzer. Neutral detergent fiber (NDF) and acid detergent fiber (ADF) were determined according to methods described by Van Soest et al. [15] and the Association of Official Analytical Chemists (method 973.18 C; [16]), respectively.

The weights of offered feed and feed refusals were measured weekly, and then feed intake was calculated on a dry-matter basis. Lambs were weighed, using an electronic scale, before morning feeding at 07:30 on day 1 of the study and every two weeks thereafter until the end of the study. Gain-to-feed ratio (feed efficiency) for each animal was calculated and expressed as bodyweight gain per kg of dry matter intake.

### 2.3. Blood Collection

Blood samples (10 mL) were collected from each lamb before the morning feeding from the jugular venipuncture on day 70. At each collection, 10 mL aliquots of blood were stored in Vacutainer tubes without additives for serum collection. Serum was obtained by centrifugation at 2400× *g* for 15 min at 4 °C and then frozen at −20 °C until further analysis. Serum concentrations of glucose, total cholesterol, total protein, albumin and globulin were analyzed using commercial kits (Randox Laboratories, Antrim, UK) and a microplate reader (Multiskan EX; Thermo Fisher Scientific Inc., Waltham, MA, USA) according to the manufacturers’ instructions.

### 2.4. Digestibility Indices

On day 60, six lambs were randomly selected from each treatment and were shifted to metabolic cages (1.2 × 1.2 m^2^) for a 7-day period (4 days of adaptation period, followed by 3 days of collection period) to find the digestion of nutrients. The sieves of the metabolic cages separated urine from the feces. During the collection period (3 days), feed and water were offered and refused, and the masses of feces were measured daily at 8: 00 from each animal. Representative samples were collected during this 3-day period, pooled (5% of feed offered and refused, and 20% of feces), and then frozen at −20 °C for determination of the apparent digestibility of fiber compositions. The samples of feed offered and refused and feces were dried for 24 h at 100 °C using an oven and then ground through a 1 mm screen. These samples were analyzed for DM, ADF, NDF and acid detergent lignin (ADL) according to the method of AOAC [17].

### 2.5. Rumen Histology

After slaughter, rumen tissue from all animals was immediately thawed, washed with a phosphate buffer solution three times, and then placed on an ice bucket. Subsamples of rumen tissues (1 cm^2^) were collected and dehydrated with 4% formalin solution and ethanol for evaluating the rumen papillae morphology. The length and width of papillae were measured using a light microscope, while the papillae density (number of papillae per cm^2^ mucosa) was estimated by a video camera. The total surface of papillae per cm^2^ mucosa was calculated according to the following equation:

The total surface of papillae per cm^2^ mucosa = papillae length × papillae width × 2 × papillae density.

### 2.6. Rumen Color Determination

The color of rumen tissues was determined using a Minolta Chroma Meter (Konica Minolta, CR-400-Japan) with a CIELAB Color System for the color values (L* = value designates lightness; a* and b* = color coordinates).

### 2.7. Carcass Quality Measurements

At the termination of the study, all lambs were first deprived of feed for 16 h and thereafter slaughtered for carcass evaluation. The hot carcass and organs, including rumen, liver, kidney, heart, spleen and testicle, were collected immediately after slaughter and weighed. After 24 h of slaughter, cold carcasses (4 °C) were weighed again, to determine dressing percentage. Thereafter, carcass characteristics and meat quality measurement were determined. The back-fat thickness (6th and 10th ribs) was measured using a ruler. The *Longissimus thoracis* muscle (12th rib) area was measured using a planimeter, and the meat color of this area was determined using a Minolta Chroma Meter (Konica Minolta, CR-400-Japan) with a CIELAB Color System for the color values (L* = lightness, a* and b* = color coordinates).

### 2.8. Statistical Analysis

All data obtained from this study were analyzed using PRO Mixed Model [18] under Analysis of Variance (ANOVA). Means were separated by Tukey Test at probability value of 0.05.

## 3. Results

The effects of dietary inclusion of SFH on the growth performance of lambs are shown in Table 2. Body weight change and average daily gain (ADG) were significantly (*p* < 0.05) higher in SFH15 compared to the control group. The initial body weight, final body weight, dry matter intake (DMI) and G:F ratio were not different (*p* > 0.05) between the control and treatment groups.

The effects of the addition of SFH on fiber digestibility and excretion in growing lambs are presented in Table 3. Dry matter intake was significantly (*p* < 0.05) lower in SFH5 and SFH10 compared to SFH15 and the control. Neutral detergent fiber intake was significantly (*p* < 0.05) lower in SFH10 compared to SFH15. However, fecal output was significantly (*p* < 0.05) higher in SFH15 compared to the treatments and the control group. Acid detergent fiber intake was significantly (*p* < 0.05) higher in SFH15 compared to SFH10.

The variations among dietary treatments in the carcass characteristics and meat quality measured in this study are presented in Table 4. Most of the carcass characteristics did not differ (*p* > 0.05) between the dietary treatments. Stomach weight increased significantly (*p* < 0.05) in SFH15 compared to the other treatments and the control group (Table 4).

Meat quality characteristics of growing lambs fed different experimental rations are given in Table 5. Cooking loss decreased significantly (*p* < 0.05) in SFH15 compared to the control group. All other characteristics did not change between the control and SFH supplemented lambs.

The effects of supplementation of SFH on blood variables in growing lambs are presented in Table 6. The total blood cholesterol and protein concentration decreased significantly (*p* < 0.05) in SFH15 compared to the control.

The ruminal color values and morphology of growing lambs that were fed the experimental diets containing different levels of SFH are given in Table 7. The rumen lightness (L*) and yellowness (b*) increased significantly (*p* < 0.05) in SFH15 compared to the control.

## 4. Discussion

In the present study, we found that supplementation of SFH15 in the TMR diet of lambs improved weight gain compared to the control. In addition, the dark color of the rumen was decreased in SFH15-supplemented lambs. This study is important since the changing rumen coloration in response to changing the feeding regimen can exploit beneficial applications. Rumen is composed of different compartments and diet-induced alterations in rumen mucosal pigmentation. It is well known that the TMR diet induces discoloration of the rumen papillae and causes acidosis, while supplementation at a certain level reduces the chances of acidosis and discoloration of the rumen. No study exists for the comparison of feeding SFH and the response parameters of the current study in the published literature. In the current study, production performance was improved with SFH15. Similar observations were also recorded by Papi et al. [19], Blanco et al. [20] and Alhidary et al. [21] reporting improved weight gain in growing lambs fed with a TMR diet plus roughage as a source of NDF. The poor performance in the control group is associated with keratinization in the rumen, which led to the decreased absorption of volatile fatty acids and might have resulted in poor weight gain in the respective groups. To a certain extent, for proper structure and function of the rumen, addition of a fiber source is very important in the TMR diet [22]. In the present study, it is inferred that the addition of SFH increased the ADG and subsequently final weight gain in the lambs. The previous reports concluded that consumption of roughage supports rumen motility and muscular development, and encourages rumination in lambs [21,23]. It has been documented that fiber intake increases rumen fill and decreases rumen acidosis. Furthermore, addition of roughage encourages chewing, saliva secretion and buffering capacity of the rumen [24,25]. From this study, it can be inferred that SFH up to 15% could be supplemented in Awassi lambs fed with the TMR diet.

Lambs supplemented with SFH at the rate of 15% (SFH15) had significantly greater DM, NDF and ADF intake than others in the group. In turn, the SFH15 showed increasing NDF and ADF in fecal output but decreases in digestibility of ADF and ADL values. This may be attributed to several possible reasons; one of the most common is the consequence of increased feed intake. These results caused a significant increase in NDF and ADF excretion and reductions in digestibility in those measurements on treatment groups compared with lambs in the control group. NDF digestibility of forage is critical for effective ruminant feeding [25,26]. In addition, the ADL for SFH15 was lower, which indicates that the lambs in this group had the best capability of nutrient digestibility of SFH as indicated by Cardoso-Gutiérrez et al. [27]. In the current study, the empty stomach weight was significantly higher in SFH15 compared to all other treatments and the control. Empty stomach weight was also higher in growing lambs supplemented with alfalfa in the TMR diet [28]. Consumption of roughage improves muscular development, supports rumen motility and increases their size [22,23]. The results indicated that experimental diets containing SFH15 decreased significantly the concentrations of total cholesterol and protein in blood. The reason for lower blood cholesterol and protein may be due to the lower intake of the TMR diet and higher SFH in the diet supplemented with SFH15.

In the current study, cooking loss decreased significantly with SFH15 compared to the control. Meat juiciness depends upon the water content [28]. Cooking loss is one of the keys to an indication of meat quality, and it has a close association with WHC. The value of WHC in meet is affected by mmobilization of water within the myofibrillar tissues of meat [29]. In the current study, lightness and yellowness of rumen decreased significantly in the rumen with SFH15. Usually, rumen color becomes darkened when fed a concentrated diet, which is associated with hardening of the mucosa [19]. Similar observations were also reported by d Alvarez-Rodríguez et al. [23] and Alhidary et al. [21]. Several theories have been forwarded for the dark color rumen of the lambs fed with TMR. Hamada et al. [30] and Alhidary et al. [20] concluded that the dark color of the rumen is possibly due to the deposition of high concentration of iron in the rumen. Blanco et al. [19] speculated that the lower rumen pH caused by a concentrated diet is responsible for the dark color of the rumen.

## 5. Conclusions

We concluded that SFH up to 15% improved weight gain, rumen color and most of the nutrient digestibility indices in Awassi lambs with no negative effects on the studied parameters.

## Figures and Tables

**Table 1 vetsci-08-00174-t001:** Ingredients and chemical composition of the experimental diets ^1^.

Item	Dietary Treatments			
	CON	SFH_5_	SFH_10_	SFH_15_
Ingredients (% of dietary dry matter)				
Barley, grain	18.6	18.6	18.6	18.6
Corn	29.1	29.1	29.1	29.1
Palm Kernel Meal	18.0	18.0	18.0	18.0
Wheat Straw	15.0	10.0	5.0	0.0
Sunflower hulls	0.0	5.0	10.0	15.0
Wheat Bran	5.2	5.2	5.2	5.2
Alfalfa hay	6.3	6.3	6.3	6.3
Salt	0.47	0.47	0.47	0.47
Limestone	2.58	2.58	2.58	2.58
Molasses	4.60	4.60	4.60	4.60
Commercial Premix ^2^	0.15	0.15	0.15	0.15
Nutrient composition, dry matter basis				
Dry matter (%)	92.43	92.21	92.06	92.12
Crude protein (%)	12.87	12.92	13.02	12.98
Crude fibre (%)	11.98	12.13	12.18	12.23
Ether extract (%)	5.33	5.18	4.91	4.98
Neutral detergent fibre (%)	32.57	33.45	33.76	34.43
Acid detergent fibre (%)	17.76	18.36	17.89	18.16
Ash (%)	8.01	8.06	8.07	8.12
Gross energy (Mcal/kg)	2.79	2.78	2.77	2.79

^1^ The experimental diets were (1) CON = the basal complete pelleted diet as control; (2) SFH_5_ = the diet supplemented with sunflower hulls at level of 5%; (3) SFH_10_ = the diet supplemented with sunflower hulls at level of 10%; and (4) SFH_15_ = the diet supplemented with sunflower hulls at level of 15%. ^2^ Contained per kg, 10,000 IU vitamin A, 1000 IU vitamin D, 20 IU vitamin E, 300 mg Mg, 24 mg Cu, 0.6 mg Co, 1.2 mg I, 60 mg Mn, 0.3 mg Se, 60 mg Zn.

**Table 2 vetsci-08-00174-t002:** Productive performance of growing Awassi lambs fed the experimental diets containing different levels of sunflower hulls.

Variables	Dietary Treatments ^1^				SE
	CON	SFH_5_	SFH_10_	SFH_15_	
Initial BW, kg	29.28	29.58	29.53	28.15	1.46
Final BW, kg	47.56	48.71	48.43	49.45	2.14
BW change, kg	18.28 ^b^	19.13 ^a,b^	18.90 ^a,b^	21.30 ^a^	1.67
ADG, g/d	0.26 ^b^	0.27 ^a,b^	0.27 ^a,b^	0.30 ^a^	0.21
DMI, kg/d	1.35	1.53	1.48	1.59	0.15
G:F ratio	0.19	0.18	0.18	0.19	0.04

^a–b^ Mean within a row, means without a common superscript differ (*p* < 0.05). ^1^ Values are for growing lambs (*n* = 48); CON, a complete pelleted diet as control diet; SFH_5_, the basal diet supplemented with sunflower hulls at level of 5%; SFH_10_, the diet supplemented with sunflower hulls at level of 10%; and SFH_15_, the diet supplemented with sunflower hulls at level of 15%.

**Table 3 vetsci-08-00174-t003:** Fiber digestibility, excretion, and balance of growing lambs fed the experimental diets containing different levels of sunflower hulls.

Variables	Dietary Treatments ^1^				SE
	CON	SFH_5_	SFH_10_	SFH_15_	
DM					
Intake, kg/d	1.53 ^a^	1.46 ^b^	1.36 ^b^	1.75 ^a^	0.22
Fecal output, kg/d	0.71	0.64	0.62	0.77	0.16
Digestibility, %	53.28	55.96	54.41	56.08	2.67
NDF					
Intake, g/d	394 ^a,b^	376 ^a,b^	348 ^b^	449 ^a^	65.0
Fecal output, g/d	88 ^b^	81 ^b^	72 ^b^	117 ^a^	25.0
Digestibility, %	77.7	78.5	79.3	73.9	9.53
ADF					
Intake, g/d	157 ^a,b^	149 ^a,b^	138 ^b^	178 ^a^	18.0
Fecal output, g/d	46 ^b^	42 ^b,c^	38 ^c^	61 ^a^	23.0
Digestibility, %	70.7 ^a^	71.8 ^a^	72.5 ^a,b^	65.7 ^b^	4.34
ADL					
Intake, g/d	19	18	17	22	2.7
Fecal output, g/d	11	10	9	13	3.2
Digestibility, %	42.1 ^a,b^	44.4 ^a,b^	47.1 ^a^	40.9 ^b^	4.1

^a–c^ Mean within a row, means without a common superscript differ (*p* < 0.05). ^1^ Values are for growing lambs (*n* = 36); CON, a basal diet as a Control; SFH_5_, the diet supplemented with sunflower hulls at level of 5%; SFH_10_, the diet supplemented with sunflower hulls at level of 10%; and SFH_15_, diet supplemented with sunflower hulls at level of 15%.

**Table 4 vetsci-08-00174-t004:** Carcass characteristics of growing Awassi lambs fed the experimental diets containing different levels of sunflower hulls.

Variables	Dietary Treatments ^1^				SE
	CON	SFH_5_	SFH_10_	SFH_15_	
Carcass profile					
Slaughter BW, kg	47.10	49.61	46.51	49.13	2.11
Empty BW, kg	43.35	45.18	42.92	44.74	1.97
Hot carcass, kg	24.20	25.03	23.95	24.55	1.16
Cold carcass, kg	23.76	24.53	23.50	24.04	1.15
Dressing^2^, %	51.28	50.47	51.51	49.98	0.82
Chilling losses, %	1.83	2.03	1.94	2.06	0.21
Organs weight, kg					
Liver	2.96	2.88	2.72	2.90	0.15
Kidneys	0.50	0.49	0.49	0.52	0.02
Heart	0.69	0.70	0.69	0.73	0.03
Stomach	4.79 ^b^	4.78 ^b^	4.42 ^b^	5.04 ^a^	0.30
Tail	10.84	15.10	12.7	11.63	0.95
Meat quality					
back fat, mm	4.75	6.26	5.16	5.83	0.64
Body wall fat, mm	5.95	6.38	6.15	6.28	0.73
Area 12th rib, cm	40.76	43.82	41.93	42.05	2.76
color components					
L*	36.35	38.12	39.5	37.51	1.25
a*	18.93	18.8	20.44	18.94	0.51
b*	8.19	8.03	7.98	7.99	0.55

^a,b^ Mean within a row, means without a common superscript differ (*p* < 0.05). ^1^ Values are for growing lambs (*n* = 48); CON, a complete pelleted diet as control; SFH_5_, the diet supplemented with sunflower hulls at level of 5%; SFH_10_, the diet supplemented with sunflower hulls at level of 10%; and SFH_15_, the diet supplemented with sunflower hulls at level of 15%. L*, lightness; a*, redness; b*, yellowness.

**Table 5 vetsci-08-00174-t005:** Meat quality of growing Awassi lambs fed the experimental diets containing different levels of sunflower hulls.

Variables	Dietary Treatments ^1^				SE
	CON	SFH_5_	SFH_10_	SFH_15_	
Visceral Depot Fat					
Pericardial fat, %	0.37	0.45	0.4	0.41	0.05
KKCF, %	1.8	2.43	2.11	1.81	0.31
Mesentery fat, %	2.08	1.71	1.93	1.51	0.22
Omental fat, %	3.06	3.3	3.15	2.5	0.41
Texture Profile Analysis					
CL %	45.66 ^a^	41.23 ^a,b^	40.73 ^a,b^	36.72 ^b^	1.92
WHC %	33.12	33.17	32.10	33.92	1.29
Shear Force (N)	39.96	37.02	32.75	30.99	2.93
MFI	110.70	95.77	92.88	79.23	7.10
Hardness (N)	5.39	4.69	3.52	4.00	0.58
Springiness	0.75	0.75	0.74	0.76	0.03
Cohesiveness	0.42	0.45	0.46	0.43	0.01
Chewiness	1.84	1.64	1.19	1.37	0.21

^a,b^ Means within a row, means without a common superscript differ (*p* < 0.05). ^1^ Values are for growing lambs (*n* = 48); CON, a complete pelleted diet as control; SFH_5_, the diet supplemented with sunflower hulls at level of 5%; SFH_10_, the diet supplemented with sunflower hulls at level of 10%; and SFH_15_, the diet supplemented with sunflower hulls at level of 15%. CL, Cooking Loss; KKCF, kidney knobs channel fat; MFI, myofibril fragmentation index and WHC, water-holding capacity.

**Table 6 vetsci-08-00174-t006:** Blood variables of growing Awassi lambs fed the experimental diets containing different levels of sunflower hulls.

Variables	Dietary Treatments ^1^				SE
	CON	SFH_5_	SFH_10_	SFH_15_	
Glucose, Mm	3.89	4.03	3.78	3.85	0.23
Total Cholesterol, Mm	1.93 ^b^	2.39 ^a^	2.06 ^a,b^	1.75 ^c^	0.12
Total Protein, g/L	76.13 ^a^	63.10 ^b^	63.40 ^b^	62.90 ^b^	6.13
Albumin, g/L	37.17	38.13	35.77	36.37	1.76
Globulin, g/L	29.93	24.97	27.60	26.53	2.96

^a,b^ Means within a row, means without a common superscript differ (*p* < 0.05). ^1^ Values are for growing lambs (*n* = 48); CON, a complete pelleted diet as control; SFH5, the diet supplemented with sunflower hulls at level of 5%; SFH10, the diet supplemented with sunflower hulls at level of 10%; and SFH15, the diet supplemented with sunflower hulls at level of 15%.

**Table 7 vetsci-08-00174-t007:** The ruminal color values and morphology of growing lambs fed the experimental diets containing different levels of sunflower hulls.

Variables ^2^	Dietary Treatments ^1^				SE
	CON	SFH_5_	SFH_10_	SFH_15_	
Color components					
L*	31.18 ^b^	31.14 ^b^	35.79 ^a,b^	40.20 ^a^	2.05
a*	4.89	4.49	4.97	5.1	0.47
b*	10.80 ^b^	10.17 ^b^	11.31 ^a,b^	13.16 ^a^	0.68
Ruminal morphologic characteristics					
PL, mm	4.1	4.38	3.79	4.52	0.27
PW, mm	0.45	0.42	0.43	0.43	0.03
PSA, cm^2^	5.89	5.79	5.26	5.96	0.51
PD, n/cm^2^	55.72	56.96	65.18	56.88	4.20
TPS, mm^2^/cm^2^	196.9	196.0	199.3	202.2	12.60

^a,b^ Means within a row, means without a common superscript differ (*p* < 0.05). ^1^ Values are for growing lambs (*n* = 48); CON, a complete pelleted diet as control; SFH5, the diet supplemented with sunflower hulls at level of 5%; SFH10, the diet supplemented with sunflower hulls at level of 10%; and SFH15, the diet supplemented with sunflower hulls at level of 15%. ^2^ PL, papillae length; PW, papillae width; PSA, papillae surface area; PD, papillae density; TSP, total surface of papillae; L*, lightness; a*, redness; b*, yellowness.

## Data Availability

The data presented in this study are available on reasonable request from the corresponding author.

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
