# Peer review of "Effects of Sunflower Hulls on Productive Performance, Digestibility Indices and Rumen Morphology of Growing Awassi Lambs Fed with Total Mixed Rations"

_vetsci, 2021, doi:10.3390/vetsci8090174_

Round 1

Reviewer 1 Report

Manuscript vetsci-1334591, entitled “Effects of Sunflower Hulls Supplementation on Productive Performance, Fiber Digestibility and Rumen Morphology of Growing Lambs”

Recommendation:       The above paper is not suitable for publication in its present form.

General Comments:

  • This article provides useful information about the effects of sunflower hulls supplementation on productive performance, fiber digestibility and rumen morphology of growing lambs. However, there are a lot of grammar, stylistic and syntax errors. In some cases, these errors negatively influence the understanding of the text.
  • Please provide a Table with the composition and analysis of the complete pelleted feed. SFH was added at the expense of the roughage or not?
  • How were the lambs that were removed into the metabolism cages selected?
  • Please check the numbering of the Tables and correct it.
  • Please define stomach weight? Rumen weight?
  • Results shown in Table 6 are not discussed in text.
  • L245-247: Where are these data presented?
  • L254-257: This is not an acceptable explanation. Please clarify.

Specific Comments:

L13: “…hulls dietary supplementation on productive…”

L14-15: “…were randomly allocated into 4 groups with 3 replicates of 4 lambs each…”

L17-18: Please delete “a basal diet supplemented sunflower 17 at level of 5% (SFH5)” (repetition)

L22: Higher in SFH15 compared to?

L26: decreased or increased?

L36: “…for ruminants [4].”

L38: What kind of effects? Positive or negative?

L40: “addition” instead of “was added”

L45: “obtained” instead of “achieved”

L50: “However” instead of “Conversely”

L52-56: Please rephrase

L57: “enhancement” instead of “variations”

L58: “…purpose of the present study…”

L75: “…were randomly allocated into 4 groups with 3 replicates of 4 lambs each…”

L76: “…offered the same…”

L106: “stored” instead of “taken”

L114-115: Please delete “for use”

L116: “from” instead of “form”

L126: “collected” instead of “taken”

L140: “…rumen, liver, kidneys, heart, spleen and testicles…”

L142: “…to determine dressing percentage. Thereafter…”

L152: Repeated measures were also applied for carcass traits and rumen characteristics?

L156: “The effects of dietary inclusion of…”

L157: “…body weight change and…”

L158: “…to the control group. The initial…”

L159: “…G:F ratio were not different between…”

L160: What do you mean by “balance”?

L163-164: Please delete (repetition)

L168-169: “The variations between dietary treatments in the carcass traits and meat quality characteristics measured in the presented study are presented in Tables 3 and 4.”

L170: “among” instead of “between”

L220-223: Please remove at the end of discussion, since the results regarding rumen are presented at the last Table.

L224: Comparison of what?

L228: “kertinization”?

L228: “decreased” instead of “less”

L227-229: Please rephrase

L234-236: Please rephrase

L238: “Lambs dietary supplemented with sunflower hulls at the level of 15% (SFH15)…”

L245-246: “…groups had improved fiber digestibility compared with control for all examined variables.”

L260: Please delete “of”

L261: “is affected by mobilization” instead of “in meet is activated by the immobilization”

L263: Please rephrase

L275: Digestibility rates were increased or decreased?

Author Response

Dear reviewer

Thank you very much for your comments on our paper. The comments are highly positive and have helped us to improve the quality of our paper. The point by point response to each comment is given below. Hope that the revised version of our paper will be acceptable to you. 

General Comments:

  • Please provide a Table with the composition and analysis of the complete pelleted feed. SFH was added at the expense of the roughage or not?

Response: Table was added

  • How were the lambs that were removed into the metabolism cages selected?

Response: added the information as track changes

  • Please check the numbering of the Tables and correct it.

Response: corrected

  • Please define stomach weight? Rumen weight?

Response: Stomach weight was comprised of all the four compartments.

  • Results shown in Table 6 are not discussed in text.

Response: The discussion of these parameters (Now Table 7) are shown in yellow color text.

  • L245-247: Where are these data presented?

Response: yes, this data is given in Table 3.

  • L254-257: This is not an acceptable explanation. Please clarify.

Response:  This is the only plausible explanation. No study exists to give comparison or provide explanations. Therefore, we provide the speculation, the exact reason how the protein and cholesterol were decreased needed to be further investigated.

Specific Comments:

L13: “…hulls dietary supplementation on productive…”

Response: corrected

L14-15: “…were randomly allocated into 4 groups with 3 replicates of 4 lambs each…”

Response: corrected

L17-18: Please delete “a basal diet supplemented sunflower 17 at level of 5% (SFH5)” (repetition)

Response: deleted

L22: Higher in SFH15 compared to?

Response: corrected

L26: decreased or increased?

Response: corrected

L36: “…for ruminants [4].”

Response: corrected

L38: What kind of effects? Positive or negative?

Response: Positive

L40: “addition” instead of “was added”

Response: added

L45: “obtained” instead of “achieved”

Response: replaced

L50: “However” instead of “Conversely”

Response: changed

L52-56: Please rephrase

Response: rephrased

L57: “enhancement” instead of “variations”

Response: changed

L58: “…purpose of the present study…”

Response: corrected

L75: “…were randomly allocated into 4 groups with 3 replicates of 4 lambs each…”

Response: revised

L76: “…offered the same…”

Response: revised

L106: “stored” instead of “taken”

Response: changed

L114-115: Please delete “for use”

Response: corrected

L116: “from” instead of “form”

Response: corrected

L126: “collected” instead of “taken”

Response: Replaced

L140: “…rumen, liver, kidneys, heart, spleen and testicles…”

Response: corrected

L142: “…to determine dressing percentage. Thereafter…”

Response: Revised

L152: Repeated measures were also applied for carcass traits and rumen characteristics?

Response: Yes,

L156: “The effects of dietary inclusion of…”

Response: corrected

L157: “…body weight change and…”

Response: changed

L158: “…to the control group. The initial…”

Response: changed

L159: “…G:F ratio were not different between…”

Response: changed

L160: What do you mean by “balance”?

Response: Balance was removed.

L163-164: Please delete (repetition)

Response: removed

L168-169: “The variations between dietary treatments in the carcass traits and meat quality characteristics measured in the presented study are presented in Tables 3 and 4.”

Response: Table numbers were revised.

L170: “among” instead of “between”

Response: changed

L220-223: Please remove at the end of discussion, since the results regarding rumen are presented at the last Table.

Response: changed

L224: Comparison of what?

Response: for comparison of feeding sunflower hulls and the response parameters.

L228: “kertinization”?

Response: corrected

L228: “decreased” instead of “less”

Response: Replaced

L227-229: Please rephrase

Response: rephrased

L234-236: Please rephrase

Response: revised

L238: “Lambs dietary supplemented with sunflower hulls at the level of 15% (SFH15)…”

Response: revised

L245-246: “…groups had improved fiber digestibility compared with control for all examined variables.”

Response: revised

L260: Please delete “of”

Response: deleted

L261: “is affected by mobilization” instead of “in meet is activated by the immobilization”

Response: corrected

L263: Please rephrase

Response: deleted this sentence, since it is irrelevant

L275: Digestibility rates were increased or decreased?

Response: most of the digestability parameters were improved.

Reviewer 2 Report

Manuscript-ID vetsci-1334591

Effects of Sunflower Hulls Supplementation on Productive Performance, Fiber Digestibility and Rumen Morphology of Growing Lambs

General remarks

Dear Authors, I have revised the abovementioned manuscript. The study of the study is very topical, as it is part of the theme of the reuse of agricultural by-products, highlighting the effective opportunity to use sunflower hulls in ruminant nutrition and providing useful dose-response results on hulls-supplemented lambs performances. Therefore, the authors' contribution to literature is appreciated. Nevertheless, in my opinion, the manuscript needs to be carefully revised in several parts. My suggestions are detailed below, section by section. Hoping to have contributed to improving the manuscript quality, I take this opportunity to wish you good work.

Sincerely

Specific comments

ABSTRACT

Authors are asked to check if the length of the abstract falls within the maximum 200 words required by the journal (I suspect there are more). Thanks

L 12-15: see a comment at L 74-76.

L 15: please, delete the comma after “each” (…12 groups of 4 lambs each, and housed….). Thanks

L 15-19: in my opinion, the description of diet treatments could be simplified, also to reduce the number of words. In any case, I suggest deleting the uppercase along with the number list [1) "a" complete pelleted]. Thanks.

L 20 (and along with the text): authors are recommended to check that the P-value is correctly indicated, according to the journal standards. It should be p (in italics) followed by spaces. I ask to verify. Thanks.

KEYWORDS

Authors are invited to make keywords consistent, using only upper or lower case. In any case, please check the template. Thanks.

I would suggest adding “digestibility” to keywords, replacing "growth" with "productive performance", and changing “carcass” as “carcass traits” and “ruminal” as “rumen”. Also, I would change the order of the keywords as follows: agricultural by-products (generic but very effective term to make the manuscript more visible), sunflower hulls, lambs, digestibility, rumen morphology, carcass traits.

I apologize to the authors for my pedantic approach, but I believe the keywords must not only clearly communicate the contents of the manuscript but also make it more attractive for a potential user.

INTRODUCTION

L 31-32: recovery and valorization of agro-industrial residues are currently indicated as key factors for the development of the circular economy models and to promote the environmental sustainability of production systems. International scientific opinion is very interested in these issues, so I would suggest to the authors, before mentioning the by-products studied, to recall them briefly. In this regard, I suggest using manuscript doi:10.3390/ani9110918 as a template (especially the introduction), which I strongly recommend citing in the references. Thanks.

L 33-35: does the reported SFH proximate composition refer to the fed-basis or dry matter? Please specify, thanks.

L 39: please, delete the comma after [5]. Thanks.

L 40: in my opinion, I would delete the weight and standard deviation values used to describe the effect of FSH dietary supplementation on sheep. I think it is too specific a concept, not very useful for a usable reading of the introduction.

L 43: please replace “upto” with “up to”. Thanks.

L 46: please delete the doubled articles (of) before cis-9. Thanks.

L 50: please replace “researchers” with “research” or, simply, “few data”. Thanks.

L 50-56: in my opinion, the sentence should be rearranged as it is unclear. I do not understand well the concept of " invariable proportions". Sorry

L 58: I notice too much space before "Therefore".

MATERIALS AND METHODS

L 66-68 (and along with the text): authors are invited to carefully check the space between the words. Thanks.

L 74-76: The authors are called upon to clarify the experimental design well. As it is written, it seems that there are 4 dietary treatments for as many groups of animals, which are 12 (12 groups of 4 lambs, in total 48 animals). The fact that 3 replicates of 4 animals exist for each treatment is implied, but not evident. Please clarify this properly. Thanks.

L 76: if available, it would be advisable for the authors to report the average temperature and humidity that occurred during the study period, perhaps reporting the corresponding value of the Temperature Humidity Index. Thanks.

L 77: in my opinion, the authors should make a table in which to summarize the detailed chemical composition of the pelleted feed and the FSH. This information is essential for evaluating the dietary choices made by the authors.

L 77-79: concise language is always advisable. However, in this case, too much information is provided in a single period at the expense of clarity. Authors are asked to paraphrase, thanks. Moreover, the bracket at end of the sentence must be added.

L 81: see a comment at L 15-19.

L 82: in my opinion, authors should better specify the methods of supplementation. The SFH were mixed with the pelleted feed, were supplied separately (and, if so, at the same time as the pellet?) or as top-dressing? Please clarify, thanks.

L 85: please replacing “evaluation” with “evaluate”. Thanks.

L 90-92: authors are invited to point out the bibliographic sources to which the methods indicated refer. Thanks.

L 98-100: in my opinion, "small animal scale for lambs" is quite redundant. I suggest using one or the other term. Furthermore, the indication of the type of scale could be more advantageously placed at the end of the sentence. Please, rearrange the sentence. Thanks.

L 113-121 (and along with the text): please, replace “faeces” with “feces”, thanks. Additionally, I believe what has been described should be supported by appropriate bibliographic references. Thanks.

L 113. to better understand if the complete separation of excreta (feces and urine) was made, the authors are asked to define the type of metabolic cage used, describing it if necessary. Thanks.

L 125: what buffer solution was used? Authors are requested to specify, thanks.

L 126 (and along with the text): authors are invited to check that the units of measurement are correctly reported, thanks.

L 140: please, add a comma after “spleens”, thanks.

L 142: please, add a comma before “again”, thanks.

STATISTIC ANALYSIS

Since the analysis of variance was performed for repeated measures, which factors were considered as non-repeated, and which were repeated? Have the iterations between the factors been evaluated? Authors are requested to specify, reporting the interactions between the factors in the results if necessary. Thanks

RESULTS

L 156-174: see a comment at L 20, thanks.

L156-157: Has table 1 been omitted? Please carefully check the tables’ order and redefining it along with the text accordingly. Thanks.

L 157: please, delete “the results showed”, thanks.

L 172-174: serum metabolic profiles were determined at 1, 28, and 70 days of the experimental period. In this paper, only the metabolites of the experimental group and the control groups were compared. Generally, the metabolites at different growth stages have a certain trend of change, and this study did not discuss whether the metabolites at the first, 28th, and 70th days changed. In addition, a time x treatment interaction can occur. Authors are requested to specify, highlighting the possible interactions along with the text. Thanks.

L 173-174: significance should be supported by the p-value magnitude.

L 177 (and other tables): please check the formatting of the tables' footnotes. Thanks.

L 189 (Table 3): I think it is also useful to report the total weight of all the organs, adding a special line.

DISCUSSION

In my opinion, the discussions should be rewritten by paying more attention to the true effects of the feed treatment tested. Throughout the paragraph little emphasis is given to the development of forestomach and their incidence on the final live weight (of considerable importance in terms of carcass yield), while the pigmentation of the mucosa is considered almost exclusively. Aspects relating to the metabolic profile, carcass traits, and meat quality (for which there are differences between the treatments) are completely ignored. No mention is made of the need to enhance a by-product, such as the FSH, although this represents, in my opinion, the foundation of the study. The authors could try to rearrange the conclusions considering these aspects. Other detailed observations are listed below.

L 219 and 231: see a comment at L 82. No mention has been made of TMR previously. In any case, I do not think it is very correct to define the mixing between pellets and FSH as TMR.

L 224: as it stands, I don't think the statement is correct. Diet-induced alterations in rumen mucosal pigmentation are well documented in the literature. Therefore, the authors are invited to correct the sentence, carefully arguing their results with those obtained in other studies. Thanks.

L 225: I invite authors to use the uppercase for “papi”, thanks.

L 227: please delete the point between “TMR diet” and “in growing lambs”.

L 233 (and along with the text): please replace “rumin” with “rumen”, please.

L 238: “subjected” seem to be less appropriate. Use “fed”, thanks.

L 270: please replace “as result of” with “because of”, thanks.

L 271-272: the authors do not have enough information to be able to state this. I suggest deleting this sentence, which appears extremely speculative.

CONCLUSIONS

In my opinion, the conclusions are only partially correct. Indeed, as the dietary level of SFH increases the weight of the viscera increases, while the weight of the carcasses was not affected. Authors are requested to support conclusions solely with the evidence they obtained from the study. I believe it more appropriate that the authors underline in the conclusion the possibility of use FSH up to levels of 15% in the diet (which has a lot of value in a circular economy perspective) without adverse effects for the lambs rather than speculating on a result which is only partial.

Author Response

General remarks

Dear Authors, I have revised the abovementioned manuscript. The study of the study is very topical, as it is part of the theme of the reuse of agricultural by-products, highlighting the effective opportunity to use sunflower hulls in ruminant nutrition and providing useful dose-response results on hulls-supplemented lambs performances. Therefore, the authors' contribution to literature is appreciated. Nevertheless, in my opinion, the manuscript needs to be carefully revised in several parts. My suggestions are detailed below, section by section. Hoping to have contributed to improving the manuscript quality, I take this opportunity to wish you good work.

Dear reviewer

Thank you very much for reviewing our paper. The comments are highly positive and have greatly improved our paper. We have revised the paper considering these comments. Hope that the revised paper will be acceptable to you.

Sincerely

Specific comments

ABSTRACT

Authors are asked to check if the length of the abstract falls within the maximum 200 words required by the journal (I suspect there are more). Thanks

Response: we have revised the abstract with main theme, however, the journal has no restriction towards the number of words in abstract…..thanks

L 12-15: see a comment at L 74-76.

Response: done.. thanks

L 15: please, delete the comma after “each” (…12 groups of 4 lambs each, and housed….). Thanks

Response: Removed…. thanks

L 15-19: in my opinion, the description of diet treatments could be simplified, also to reduce the number of words. In any case, I suggest deleting the uppercase along with the number list [1) "a" complete pelleted]. Thanks.

Response: Simplified….thanks

L 20 (and along with the text): authors are recommended to check that the P-value is correctly indicated, according to the journal standards. It should be p (in italics) followed by spaces. I ask to verify. Thanks.

Response: corrected…thanks

KEYWORDS

Authors are invited to make keywords consistent, using only upper or lower case. In any case, please check the template. Thanks.

Response: formatted according to the journal style..thanks

I would suggest adding “digestibility” to keywords, replacing "growth" with "productive performance", and changing “carcass” as “carcass traits” and “ruminal” as “rumen”. Also, I would change the order of the keywords as follows: agricultural by-products (generic but very effective term to make the manuscript more visible), sunflower hulls, lambs, digestibility, rumen morphology, carcass traits.

I apologize to the authors for my pedantic approach, but I believe the keywords must not only clearly communicate the contents of the manuscript but also make it more attractive for a potential user.

Response: we have formatted the keywords according to the suggestions and journal format.

INTRODUCTION

L 31-32: recovery and valorization of agro-industrial residues are currently indicated as key factors for the development of the circular economy models and to promote the environmental sustainability of production systems. International scientific opinion is very interested in these issues, so I would suggest to the authors, before mentioning the by-products studied, to recall them briefly. In this regard, I suggest using manuscript doi:10.3390/ani9110918 as a template (especially the introduction), which I strongly recommend citing in the references. Thanks.

Response: I have added new information as suggested…Thanks

L 33-35: does the reported SFH proximate composition refer to the fed-basis or dry matter? Please specify, thanks.

Response: dry matter basis…thanks

L 39: please, delete the comma after [5]. Thanks.

Response: Deleted…thanks

L 40: in my opinion, I would delete the weight and standard deviation values used to describe the effect of FSH dietary supplementation on sheep. I think it is too specific a concept, not very useful for a usable reading of the introduction.

Response: deleted…..thanks

L 43: please replace “upto” with “up to”. Thanks.

Response: corrected…thanks

L 46: please delete the doubled articles (of) before cis-9. Thanks.

Response: deleted…thanks

L 50: please replace “researchers” with “research” or, simply, “few data”. Thanks.

Response: done…thanks

L 50-56: in my opinion, the sentence should be rearranged as it is unclear. I do not understand well the concept of " invariable proportions". Sorry

Response: revised…thanks

L 58: I notice too much space before "Therefore".

Response: removed…thanks

MATERIALS AND METHODS

L 66-68 (and along with the text): authors are invited to carefully check the space between the words. Thanks.

Response: Corrected…thanks

L 74-76: The authors are called upon to clarify the experimental design well. As it is written, it seems that there are 4 dietary treatments for as many groups of animals, which are 12 (12 groups of 4 lambs, in total 48 animals). The fact that 3 replicates of 4 animals exist for each treatment is implied, but not evident. Please clarify this properly. Thanks.

Response: Tried to clarify…please check…thanks

L 76: if available, it would be advisable for the authors to report the average temperature and humidity that occurred during the study period, perhaps reporting the corresponding value of the Temperature Humidity Index. Thanks.

Response: Provided….thanks

L 77: in my opinion, the authors should make a table in which to summarize the detailed chemical composition of the pelleted feed and the FSH. This information is essential for evaluating the dietary choices made by the authors.

Response: provided as Table 1.

L 77-79: concise language is always advisable. However, in this case, too much information is provided in a single period at the expense of clarity. Authors are asked to paraphrase, thanks. Moreover, the bracket at end of the sentence must be added.

Response: Revised…thanks

L 81: see a comment at L 15-19.

Response: Revised …..thanks

L 82: in my opinion, authors should better specify the methods of supplementation. The SFH were mixed with the pelleted feed, were supplied separately (and, if so, at the same time as the pellet?) or as top-dressing? Please clarify, thanks.

Response: Revised…thanks

L 85: please replacing “evaluation” with “evaluate”. Thanks.

Response: revised as suggested.

L 90-92: authors are invited to point out the bibliographic sources to which the methods indicated refer. Thanks.

Response: as pointed out in [14] and [15]…thanks

L 98-100: in my opinion, "small animal scale for lambs" is quite redundant. I suggest using one or the other term. Furthermore, the indication of the type of scale could be more advantageously placed at the end of the sentence. Please, rearrange the sentence. Thanks.

Response: revised..thanks

L 113-121 (and along with the text): please, replace “faeces” with “feces”, thanks. Additionally, I believe what has been described should be supported by appropriate bibliographic references. Thanks.

Response: revised..thanks

L 113. to better understand if the complete separation of excreta (feces and urine) was made, the authors are asked to define the type of metabolic cage used, describing it if necessary. Thanks.

Response: provided ….thanks

L 125: what buffer solution was used? Authors are requested to specify, thanks.

Response: added…thanks

L 126 (and along with the text): authors are invited to check that the units of measurement are correctly reported, thanks.

Response: corrected…thanks

L 140: please, add a comma after “spleens”, thanks.

Response: added…thanks

L 142: please, add a comma before “again”, thanks.

Response: added…thanks

STATISTIC ANALYSIS

Since the analysis of variance was performed for repeated measures, which factors were considered as non-repeated, and which were repeated? Have the iterations between the factors been evaluated? Authors are requested to specify, reporting the interactions between the factors in the results if necessary. Thanks

Response: the repeated measures include blood, tissue and rumen colors. Interaction was not performed… thanks

RESULTS

L 156-174: see a comment at L 20, thanks.

Response: changed..thanks

L156-157: Has table 1 been omitted? Please carefully check the tables’ order and redefining it along with the text accordingly. Thanks.

Response: Tables were set ….thanks

L 157: please, delete “the results showed”, thanks.

Response: deleted …thanks

L 172-174: serum metabolic profiles were determined at 1, 28, and 70 days of the experimental period. In this paper, only the metabolites of the experimental group and the control groups were compared. Generally, the metabolites at different growth stages have a certain trend of change, and this study did not discuss whether the metabolites at the first, 28th, and 70th days changed. In addition, a time x treatment interaction can occur. Authors are requested to specify, highlighting the possible interactions along with the text. Thanks.

Response: we have now corrected the text in the materials and methods. The Table 6 of blood variables is the data on day 70 only. We have revised ….thanks

L 173-174: significance should be supported by the p-value magnitude.

Response: corrected…thanks

L 177 (and other tables): please check the formatting of the tables' footnotes. Thanks.

Response: Please clarify more ….we did not understand…thanks

L 189 (Table 3): I think it is also useful to report the total weight of all the organs, adding a special line.

Response: plz clarify….thanks

DISCUSSION

In my opinion, the discussions should be rewritten by paying more attention to the true effects of the feed treatment tested. Throughout the paragraph little emphasis is given to the development of forestomach and their incidence on the final live weight (of considerable importance in terms of carcass yield), while the pigmentation of the mucosa is considered almost exclusively. Aspects relating to the metabolic profile, carcass traits, and meat quality (for which there are differences between the treatments) are completely ignored. No mention is made of the need to enhance a by-product, such as the FSH, although this represents, in my opinion, the foundation of the study. The authors could try to rearrange the conclusions considering these aspects. Other detailed observations are listed below.

Response: we have improved the discussion to the best of our efforts. Any further suggestion for the improvement will be highly welcomed… thanks

L 219 and 231: see a comment at L 82. No mention has been made of TMR previously. In any case, I do not think it is very correct to define the mixing between pellets and FSH as TMR.

Response: The pellet diet was composed of TMR. We have revised it in the materials and methods.

L 224: as it stands, I don't think the statement is correct. Diet-induced alterations in rumen mucosal pigmentation are well documented in the literature. Therefore, the authors are invited to correct the sentence, carefully arguing their results with those obtained in other studies. Thanks.

Response: Changed…thanks

L 225: I invite authors to use the uppercase for “papi”, thanks.

Response: corrected… thanks

L 227: please delete the point between “TMR diet” and “in growing lambs”.

Response: removed… thanks

L 233 (and along with the text): please replace “rumin” with “rumen”, please.

Response: removed.. .thanks

L 238: “subjected” seem to be less appropriate. Use “fed”, thanks.

Response: Changed…thanks

L 270: please replace “as result of” with “because of”, thanks.

Response: changed… thanks

L 271-272: the authors do not have enough information to be able to state this. I suggest deleting this sentence, which appears extremely speculative.

Response: deleted…thanks

CONCLUSIONS

In my opinion, the conclusions are only partially correct. Indeed, as the dietary level of SFH increases the weight of the viscera increases, while the weight of the carcasses was not affected. Authors are requested to support conclusions solely with the evidence they obtained from the study. I believe it more appropriate that the authors underline in the conclusion the possibility of use FSH up to levels of 15% in the diet (which has a lot of value in a circular economy perspective) without adverse effects for the lambs rather than speculating on a result which is only partial.

Response: revised as suggested.

Round 2

Reviewer 1 Report

Although the article is substantially improved, further corrections should be made.

Why did you use "repeated measures"? Which parameters were repeatedly measured?

Please refer only to significant differences. For example, in L223-224 superscripts are the same or digestibility is decreased. As a result, we cannot say that digestibility rates were improved

Please remove the parameters that refer to meat quality from Table 4 to Table 5. At the same time, results concerning meat quality are not discussed. Please cite the related literature.

In the revised form of the manuscript, although authors have made the recommended corrections, the previous sentences remain. For example, in L15 (and 64-65), please delete "divided into 12 groups of 4 lambs each", in L18, the phrase "a basal diet supplemented sunflower at level of 5%" is still repeated, in L26 please delete "decreased", in L39 "the", in L43 "was added", L47 "achieved", L52 "conversely", L121 "taken", L151 "change", L224 "better", "studded" etc (please check specific comments of the previous review)

L135, 137: Please correct according to the comments of the previous review

L153: "balance"?

L156-157: Please delete (repetition)

L162: "...in Table 4."

L203-204: Please rephrase

Author Response

Reviewer 1

Although the article is substantially improved, further corrections should be made.

Response: thank you very much for your comments and appreciation. We are trying to follow up your comments in letter and spirit. Thanks

Why did you use "repeated measures"? Which parameters were repeatedly measured?

Response: Repeated measure were removed. Sorry for the mistake.

Please refer only to significant differences. For example, in L223-224 superscripts are the same or digestibility is decreased. As a result, we cannot say that digestibility rates were improved

Response: Corrected… thank you

Please remove the parameters that refer to meat quality from Table 4 to Table 5. At the same time, results concerning meat quality are not discussed. Please cite the related literature.

Response:

In the revised form of the manuscript, although authors have made the recommended corrections, the previous sentences remain. For example, in L15 (and 64-65), please delete "divided into 12 groups of 4 lambs each", in L18, the phrase "a basal diet supplemented sunflower at level of 5%" is still repeated, in L26 please delete "decreased", in L39 "the", in L43 "was added", L47 "achieved", L52 "conversely", L121 "taken", L151 "change", L224 "better", "studded" etc (please check specific comments of the previous review)

Response: all these changes were incorporated. Thanks

L135, 137: Please correct according to the comments of the previous review

Response: Corrected. Thanks

L153: "balance"?

Response: corrected…thanks

L156-157: Please delete (repetition)

Response: corrected…. thanks

L162: "...in Table 4."

Response: added… thanks

L203-204: Please rephrase

Response…. Corrected…. Thanks

Reviewer 2 Report

Dear authors,

I re-evaluated the manuscript, in its revised version. I thank the authors for accepting most of my recommendations. I sincerely hope they have been helpful in improving the quality of the manuscript. Coming to the review, some other clarifications and small changes would be necessary, as listed below. The comments refer to the line lines of the new version.

Thank you for your patience. I wish you good work.

Abstract: please forgive me, but the length of the summary should not exceed 200 words at the most. Check it out for yourself: https://www.mdpi.com/journal/vetsci/instructions.

L 16: in my opinion, the capital letter at the top of the list defining diet treatments is not necessary. Thanks.

L 26: please, replace “upto” in “up to”.  Thanks.

L 49: add the space between article and noun (…improved of cis-9….). Thanks.

L 49: I suggest making all fatty acids consistent, using lowercase or uppercase letters. In addition, I would delete "and" before "trans-11" and add the comma after "muscle". Thanks.

 Table 1 (and other tables): please, formatting tables’ captions and footnotes according to the journal requirements (https://www.mdpi.com/journal/vetsci/instructions), thanks.

Statistic analysis: in my opinion, the authors should rewrite this paragraph. Based on the latest indications, the blood data were not analyzed for repeated measurements, since one sampling was performed. Therefore, the authors should specify which parameters were analyzed for repeated measures and which were not. Also, if the iterations between factors have not been evaluated, the authors should make this explicit.

Author Response

I re-evaluated the manuscript, in its revised version. I thank the authors for accepting most of my recommendations. I sincerely hope they have been helpful in improving the quality of the manuscript. Coming to the review, some other clarifications and small changes would be necessary, as listed below. The comments refer to the line lines of the new version.

Thank you for your patience. I wish you good work.

Response: thank you for your encouragement.

Abstract: please forgive me, but the length of the summary should not exceed 200 words at the most. Check it out for yourself: https://www.mdpi.com/journal/vetsci/instructions.

Response: reduced to less than 200 words. Thanks for your guideline

L 16: in my opinion, the capital letter at the top of the list defining diet treatments is not necessary. Thanks.

Response: corrected. thanks

L 26: please, replace “upto” in “up to”.  Thanks.

Response: corrected. .. thanks

L 49: add the space between article and noun (…improved of cis-9….). Thanks.

Response: corrected…thanks

L 49: I suggest making all fatty acids consistent, using lowercase or uppercase letters. In addition, I would delete "and" before "trans-11" and add the comma after "muscle". Thanks.

Response: corrected …. thanks

 Table 1 (and other tables): please, formatting tables’ captions and footnotes according to the journal requirements (https://www.mdpi.com/journal/vetsci/instructions), thanks.

Response: we have done to the best of our knowledge, however, the journal composing department will help us in formatting if the paper was accepted. …. thanks

Statistic analysis: in my opinion, the authors should rewrite this paragraph. Based on the latest indications, the blood data were not analyzed for repeated measurements, since one sampling was performed. Therefore, the authors should specify which parameters were analyzed for repeated measures and which were not. Also, if the iterations between factors have not been evaluated, the authors should make this explicit.

Response: corrected….thanks